# Ontogenetic Shifts in Body Morphology of Demersal Sharks’ Species (Order: Squaliformes) Inhabiting the Western-Central Mediterranean Sea, with Implications for Their Bio-Ecological Role

**DOI:** 10.3390/biology12081150

**Published:** 2023-08-19

**Authors:** Andrea Bellodi, Antonello Mulas, Louise Daniel, Alessandro Cau, Cristina Porcu, Pierluigi Carbonara, Maria Cristina Follesa

**Affiliations:** 1Stazione Zoologica Anton Dohrn, Contrada Porticatello 29, 98167 Messina, Italy; 2Department of Life and Environmental Sciences, University of Cagliari, 09126 Cagliari, Italy; amulas@unica.it (A.M.); alessandrocau@unica.it (A.C.); cporcu@unica.it (C.P.); follesac@unica.it (M.C.F.); 3Consorzio Nazionale Interuniversitario per le Scienze Mare (CoNISMa), Piazzale Flaminio 9, 00196 Roma, Italy; 4Institut Agro Rennes Angers, Fishery Sciences, 65 Rue de Saint-Brieuc, 35042 Rennes, France; louise.daniel76@gmail.com; 5Fondazione COISPA ETS, Via dei Trulli 18/20, 70126 Bari, Italy; carbonara@coispa.it

**Keywords:** ontogenetic growth, geometric morphometric, morphology, demersal sharks, *Centrophorus uyato*, *Dalatias licha*, *Etmopterus spinax*, *Oxynotus centrina*, *Squalus blainville*

## Abstract

**Simple Summary:**

The present study intended to investigate possible variations in body proportions during the growth of five different shark species that inhabit the Sardinian coastal waters (Central-Western Mediterranean Sea). Our results indicate that all of the evaluated species, while not presenting differences between sexes, seemed to show a generally more elongated body and a wider caudal fin when fully grown. This result may indicate a shift in the role that these species play in their environment during their individual growth.

**Abstract:**

Several elasmobranch species undergo shifts in body proportions during their ontogenetic growth. Such morphological changes could reflect variation in diet, locomotion, or, more broadly, in the species’ interactions with their environment. However, to date, only a few studies have been conducted on this topic, and most of them focused on particular body regions. In the present study, the ontogenetic growth of five different demersal shark species was investigated by using both traditional linear morphometry of the entire body and shape analysis of the caudal fin. A total of 449 sharks were analysed: 95 little gulper sharks, 80 longnose spurdogs, 103 kitefin sharks, 124 velvet belly lanternsharks, and 47 angular roughsharks. From each specimen, 36 linear morphometric measurements were taken. While a first canonical analysis of principal coordinates ruled out the possibility of different growth patterns between males and females, the same analysis statistically discriminated between small and large individuals in every species based on their morphology. A Similarity Percentage analysis revealed that the most important measurements in distinguishing these two groups were those related to body lengths, indicating that large individuals are more elongated than small individuals. The shape analysis of caudal fins revealed allometric growth during ontogenetic development, with adult individuals having a wider fin (discriminant analysis, *p* < 0.05). These findings could be related to changes in predatory skills, supporting the hypothesis of a shift in the ecological role that these sharks play in their environment, thus providing new essential information for their conservation.

## 1. Introduction

Cartilaginous fish have often been recognized as the apex predator of the environments they inhabit [1,2,3], even by the general public. For this reason, and due to their ecological role in controlling the abundance of prey populations [2,4,5], they are often indicated as key species for the maintenance of ecosystem stability [4]. Consequently, stakeholders and fishery management programs are focusing their attention on these species’ interactions with the ecosystem. Nonetheless, while information on the feeding behaviour and trophic level are usually available for many species [2], little is known regarding how ontogenetic growth affects their environmental role. In this context, shifts in body proportions during growth have been observed in several species of elasmobranchs, e.g., as found in [6,7,8,9,10,11]. Since such shifts may be linked to changes in, e.g., the species’ interaction with the habitat, diet composition, or locomotion, expanding our understanding of the ontogenetic trajectories of sharks will allow for a more precise evaluation of stock conditions as well as enabling the implementation of appropriate measures for protection and management [12]. Nonetheless, to date, most studies concerning the ontogenetic variations in the body morphology of elasmobranchs have usually focused on specific body parts, particularly the head region, e.g., in [13,14,15,16], or the fins [17,18,19]. In turn, very few studies have involved an analysis of the animal’s entire body [8,20], and even in those cases, observations and interpretations are generally constrained to the head and fins. A scrutiny of the literature reveals how different scenarios have emerged from different shark species, such as how both isometric and allometric growth have been reported in addition to sex-related differences in body shape [20,21]. The ecological shifts that come with the alteration (or not) of body proportions between juveniles and adults are numerous. For example, Fu et al. [7], while demonstrating how a tiger shark’s head becomes progressively broader during growth, hypothesized that this fact could be related to the necessity of a more powerful bite in order to allow for their dietary shift towards bigger and tougher prey such as sea turtles or cetaceans. On the other hand, changes in head morphology, in some cases, have been also ascribed to mating behaviour [20,21]. Indeed, a marked sexual dimorphism was reported in the head region of the small spotted catshark [20,21].

Fins, especially the caudal fin, represent another source of morphometric differences during ontogenetic growth, and also depict a rather heterogeneous scenario. In this regard, *Squalus acanthias* (Linnaeus, 1758), *Carcharhinus limbatus* Müller & Henle, 1839, and *Ginglymostoma cirratum* (Bonnaterre, 1788) are examples of those species that exhibit isometric changes in the caudal fin as they grow [6,13]. This situation has been theorized to be more common in small to medium-sized sharks. Large apex predators, on the other hand, such as *Carcharodon carcharias* (Linnaeus, 1758), *Galeocerdo cuvier* Péron & Lesueur, 1822, and *Carcharhinus leucas* (Valencienne, 1839), appear to have a caudal fin area that scales with negative allometry [6,17].

Given these premises, the current study aims to evaluate the changes in body proportions during ontogenetic growth in several species of demersal sharks inhabiting the Mediterranean Sea. In this regard, five different species belonging to the order Squaliformes were caught all around the Sardinian Sea (FAO-GFCM Geographic sub-area 11). This area, due to its peculiar geographic position of being located in the centre of the Western Mediterranean basin, is often considered to be a hotspot for demersal shark biodiversity and abundance [22,23,24]. In addition to the velvet belly lanternshark (*Etmopterus spinax* (Linnaeus, 1758)), the longnose spurdog (*Squalus blainville* (Risso, 1827)), and the little gulper shark (*Centrophorus uyato* (Rafinesque, 1810)), which are considered common species in the area, two rare and poorly known species were also investigated: the kitefin shark (*Dalatias licha* (Bonnaterre, 1788)) and, notably, the angular roughshark (*Oxynotus centrina* (Linnaeus, 1758)) [25,26]. Furthermore, an entire body morphometrical analysis was followed by a more in-depth shape analysis of the caudal fin, yielding the most comprehensive scenario regarding the effect of changes in body proportions during ontogenetic growth on the ecological role of these important species.

## 2. Materials and Methods

### 2.1. Sampling

The specimens for each of the 5 analysed species were sampled during experimental trawl surveys, such as the Mediterranean International Trawl Survey (MEDITS) [27], or from accidental captures by commercial hauls between 2009 and 2022 at a depth ranging from 123 to 730 m around Sardinia (Central-Western Mediterranean Sea).

The total length (TL) of the specimens, defined as the distance between the snout tip and the projection of the caudal fin posterior margin when in a natural position, was taken in the laboratory and the sex and maturity stages were also determined following the maturity scales provided by [28,29]. Each specimen was photographed next to a unit of measure with a digital camera (Canon 650D, always equipped with a Canon 18–55 mm lens) that was placed perpendicularly to the animal in order to proceed with morphometric analyses.

The Individuals of each species were subdivided into two size groups for this study: “small” and “large”. The specimens were considered “small” if their TL was less than that of the smallest mature specimen observed for each species: 55 cm for *C. uyato* and *O. centrina*, 65 cm for *D. licha*, 25 cm for *E. spinax*, and 45 cm for *S. blainville*.

### 2.2. Body Morphometric Analysis

A total of 95 little gulper sharks (*C. uyato*), 103 kitefin sharks (*D. licha*), 124 velvet belly lanternsharks (*E. spinax*), 47 angular roughsharks (*O. centrina*), and 80 longnose spurdogs (*S. blainville*) were sampled. The free software TPSDig2 v.2.31 (Rohlf 2015) was used to take the measurements, all expressed in centimetres, which were defined according to [30,31] (Figure 1; Table 1). In addition, the measurements were expressed as a percentage of TL in order to eliminate the effect of the size of the individual and to be able to compare body proportions [32,33]. In particular, 36 linear morphometric measurements were taken, apart from TL, for *C. uyato*, *E. spinax*, and *O. centrina*; 34 were taken for *D. licha* as the species does not present with dorsal spines, and thus the measurements D1B’ and D2B’ (first and second dorsal fin spine to inner margin, respectively) could not be measured. Finally, 32 measurements besides TL were taken for *S. blainville* with D1B’, D2B’, P2B (pelvic fin base), and PSP (prespiracular length) missing.

To test the hypothesis that each studied species undergoes allometric growth regardless of sex during growth, a canonical analysis of principal coordinates (CAP) was performed with the PRIMER v.7 software [34] on the similarity matrices based on Euclidean distance previously computed with the specimens classified by their sex. When significant differences between sexes were not detected, the specimens of both sexes were grouped in order to analyse the effect of size on body proportions. After that the effect of sex was evaluated, another CAP was performed to test if body proportions are different between “small” and “large” specimens. A Similarity Percentage (SIMPER) [34] analysis was also performed to determine which morphometric measurements were most responsible for the definition of the size groups.

### 2.3. Geometric Morphometric Analysis of the Caudal Fin

The more traditional linear morphometric analysis was completed by carrying out a geometric morphometric analysis of the caudal fin. Pictures of the caudal fin of 70 *C. uyato,* 66 *D. licha*, 70 *E. spinax*, 47 *O. centrina*, and 70 *S. blainville* were used in order to compare the shape of the fin between the “small” and the “large” groups of individuals in each species. A TPS file containing the pictures was created for each species using tpsUtil v.1.82 [35], and landmarks were placed on each of the pictures using tpsDig2 v.2.17 and v.2.31 [35]. Landmarks were placed on easily identifiable spots on all species: 7 for *C. uyato*, *D. licha*, and *E. spinax*, 6 for *O. centrina*, and 5 for *S. blainville*, as the shape of the caudal fin is different between species (Figure 2).

The statistical analyses were realised for each species independently following the same protocol. All of the statistical tests for determining the geometric morphometrics of the caudal fin were performed using the software package MorphoJ [36]. The coordinates from the TPS file first underwent a Procrustes transformation that was aligned by using the principal axes to rotate, translate, and scale the pictures, resulting in an image that allows for the shape comparison using the landmark configurations. Similarly to the analyses with the linear morphometric measurements, the effect of “sex” was first tested in order to define the groups used for the analyses to compare the size groups. Discriminant function analyses were performed to compare the defined groups and determine if the “small” and “large” specimens have significative differences in regard to the shape of the caudal fin. A wire-framed graph was obtained which illustrates the differences between the two size groups. After generating a covariance matrix, a principal component analysis (PCA) was performed to visualize the individual variations via a bi-plot showing the size groups.

## 3. Results

### 3.1. Linear Morphometric Analysis

Five matrices with body measurements expressed as a % of TL were obtained from the morphometric analyses, with one for each of the species. Detailed information on the analysed specimens are available in Table 2.

Firstly, CAP was used for the classification of the individuals by sex for each species (Table 3). The cross validation was used to determine if significant intraspecific differences existed between males and females. High misclassification errors were found for all of the species. In particular, they were 39.81% for *D. licha*, with 22 females and 27 males misclassified; 36.84% for *C. uyato*, with 13 females and 22 males misclassified; 25.00% for *S. blainville*, with 9 females and 11 males misclassified; 33.07% for *E. spinax*, with 23 females and 18 males misclassified; and 34.04% for *O. centrina*, with 7 females and 9 males misclassified (Table 3). Therefore, the separation between sex was not clear.

Conversely, the second CAP performed for each species using the classification by size group (“small” or “large”) returned a sharper separation. In fact, all of the species presented a high percentage of correct assignments with very few misclassifications: 98.7% of *D. licha* specimens were correctly classified, with a single misclassification of a “small” individual erroneously placed among the “large” group; 93.68% for *C. uyato*, with only 6 misclassified individuals; 87.50% for *S. blainville*, with 10 misclassifications; 83.87% for *E. spinax*, with 20 wrong assignments; and 87.23% for *O. centrina*, with 6 misclassified individuals (Table 3).

The SIMPER analysis (Appendix A) enabled us to perceive which measurements contributed the most to the classification. Similitudes existed between the different species as most of the principal contributing measurements were body length, such as SVL which was the most important contributor to the differences between size groups for four out of five species (with SIMPER Contrib% = 19.20% for *C. uyato*, 16.98% for *D. licha*, 18.59% for *E. spinax*, and 19.79% for *O. centrina*) and the third most important for the last species, *S. blainville* (SIMPER Contrib% = 10.71%). In the same way, PP2, which is a similar measure to SVL, was the second most important contributor for the same four species (SIMPER Contrib% = 17.05% for *C. uyato*, 12.03% for *D. licha*, 14.81% for *E. spinax*, and 18.10% for *O. centrina*) and the sixth most important for *S. blainville* (SIMPER Contrib% = 5.34%). PD2 was the third most important for three out of five species, the second most important for *S. blainville*, and fourth most important for *E. spinax*. Other body lengths were among the top four with the greatest contribution, such as FL for *S. blainville* (SIMPER Contrib% = 18.27%), HDL for *C. uyato* (SIMPER Contrib% = 8.69%), IDS for *D. licha* (SIMPER Contrib% = 9.54%), and the tail’s length PCA for *S. blainville* (SIMPER Contrib% = 7.91%). However, two exceptions could be observed: the first was for *E. spinax* with INW (SIMPER Contrib% = 10.37%), which is a measure of a part of the head, being the third most contributing measurement, and the second was *O. centrina* with D1H (SIMPER Contrib% = 5.42%), which is a measure of a part of the dorsal fin.

### 3.2. Geometric Morphometric Analysis

The information on the specimens used for each species is available in Table 4.

Discriminant function analyses, performed for each species to test the difference between males and females, revealed no significant differences (*p*-value > 0.05) for four out of the five analysed species (Figure 3), with significant differences only detected in *O. centrina* (*p*-value = 0.0156) (Figure 3F). In this case, the shape of the females’ caudal fin appeared a bit wider compared to that of males in the wire-framed diagram (Figure 3F), highlighting a separation between the two groups.

Discriminant function analyses between the size groups was performed by grouping males and females of the same sizes together, showing statistical significantly differences in the caudal fin shape (*p* < 0.001). The wire-framed graphs (Figure 3) generally indicate that these differences may be related to a wider caudal fin for larger specimens with respect to the smaller ones.

Additionally, a PCA was then performed for each of the species (Figure 4). The PCAs showed that the first two components accounted for 73.69%, 56.84%, 63.22%, 73.93%, and 81.74% of the variance for *C. uyato*, *D. licha*, *E. spinax*, *O. centrina*, and *S. blainville*, respectively. As for the first three components, they represented 82.51%, 71.80%, 72.29%, 87.24%, and 91.16%. The plots show the separation between the two size groups for all of the species. Moreover, since a difference in the caudal fin shape between sexes was found for *O. centrina*, a second PCA was performed for this species with sex as the classifier, and a plot was obtained (Figure 4F).

## 4. Discussion

The CAP based on a somatic linear measurement failed in discriminating between males and females of the analysed species examined here, showing a rather high misclassification percentage that ranged between 22.6 and 41.3%. Similarly, the results from the caudal fin geometric morphometric analysis ruled out the possibility of sex-driven differences in regard to fin shape. The only exception to the latter statement was represented by *O. centrina*, whose caudal fin’s shape was found to be significantly different among sexes. However, this outcome might be a consequence of the lower sample number of this species in comparison to the other considered species analysed here, which is a direct consequence of its rarity [25,26]. Therefore, the results of the present paper regarding *O. centrina*, although representing a solid baseline for future studies and being a consistent step towards a better comprehension of its biology, should be taken carefully and are considered as preliminary. In general, our results appear in contrast with what was reported for other species, such as *Scyliorhinus canicula* (Linnaeus, 1758), for which differences in the male’s head region were linked to mating behaviour involving mating bites [20,21]. In this regard, the species analysed here might either not present such behaviour, even though it is commonly reported in sharks and rays [37], or, more likely, they share the same habits but in a less extremised way that does not necessitate a marked structural change in body morphology.

Conversely, the morphometric analysis carried out between size groups returned a clear separation between small and large individuals, both in terms of body proportions and caudal fin shape, of every shark species investigated. It is also worth noting that the linear measurements that were found to account for the majority of the differences (SIMPER) between size groups were mostly the same in all investigated species and were always related to body lengths (e.g., the preanal length, the distance between the snout and the ventral fins, the fork length, and the interdorsal space) (Appendix A). Therefore, these species seem to show allometric growth in adults that have bodies that become progressively more elongated during ontogeny. A similar situation has also been observed in other demersal sharks such as the zebra shark (*Stegostoma fasciatum* Hermann, 1783), for which it has been hypothesized that the progressive elongation of the body might be functional to create more room in the thoracic cavity for the development of the reproductive organs [38]. However, given the absence of significant discrepancies in linear measurements between sexes described in the present study, and also considering the higher space requirement that females’ reproductive organs (and, even embryos in viviparous species [39,40]) generally necessitate with respect to males’, the need for extra room for the reproductive organs’ development as a main cause of body elongation in the species investigated here seems unlikely to be a driving factor in their ontogenetic growth. Nonetheless, considering that it has often proven difficult to draw a conclusion on whether there are differences or not between sexes [12], this hypothesis should be investigated further before being completely rejected. In this regard, the reason for such discrepancies in the morphology should be searched for elsewhere, as it could be associated with swimming behaviour. Indeed, changes in fish body length have often been related to an increase in swimming efficiency and capacity as an adaptation for a better predation capability, thus influencing feeding habits [41], or as a means to escape from predators [42]. In this regard, the positive allometric growth of the caudal fin of the Squaliformes investigated here, which is in agreement with that reported in other Mediterranean regions for *D. licha*, *S. blainville*, and *S. canicula* [43], might represent an additional clue that points towards this latter hypothesis, as a wider caudal fin area with respect to body size could provide an extra boost, increasing the shark’s chances to successfully hunt bigger and faster nektonic organisms [43,44,45]. Consequently, the ability to feed on these kinds of prey could represent a shift in the trophic level of these species during their ontogenetic growth, thus also modifying the role they play in their ecosystems [44], a pattern which has already been described in several other elasmobranch species [2].

The positive allometry in caudal fin growth reported here represents the opposite situation of what is commonly reported for large pelagic sharks, which are often characterized by negative allometric growth in the caudal region. A proportionally smaller and frequently less heterocercal tail could be ascribed to the fact that adults migrate more than juveniles and that they need to travel longer distances to search for bigger prey while expending as little energy as possible by minimizing the lift–drag ratio [7,8,17].

On the other hand, smaller shark species are commonly reported as showing isometric growth in the caudal region, such as *Ginglymostoma cirratum* (Bonnaterre, 1788) [6] or *Squalus acanthias* (L. 1758), which is a congeneric species to *S. blainville* that, in the present study, was found to have positive allometry.

## 5. Conclusions

In conclusion, the present paper represents a first attempt to investigate the ontogenetic shift in body proportions, both by considering the linear morphometrics on the whole body and geometric morphometry on the caudal fins of several demersal sharks, including some rare species. All of the results reported here seem to delineate a general growth pattern shared by all of the considered species belonging to the order Squaliformes, according to which these sharks appear to become generally more elongated and contemporarily present with a wider caudal fin during their ontogenetic development. Considering how this distortion in morphological proportions between juveniles and adults may affect the life cycle of a species, causing, for example, changes in habitats, diet, and locomotion [44], the information reported here could contribute to unravelling new and more precise knowledge on the functional role played in the ecosystems by these animals during their life cycle.

## Figures and Tables

**Figure 1 biology-12-01150-f001:**
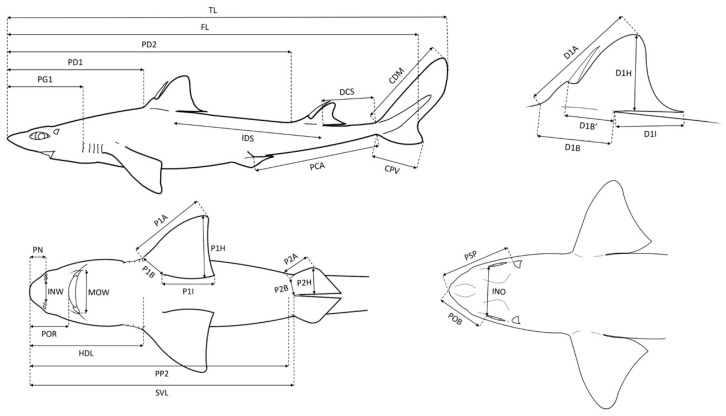
Body morphometric measurements and the relative acronyms, which are defined in Table 1.

**Figure 2 biology-12-01150-f002:**
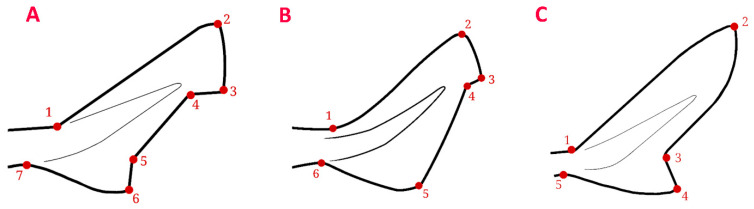
Diagram of the position of the landmarks established for *Centrophorus uyato*, *Dalatias licha* and *Etmopterus spinax* (**A**); *Oxynotus centrina* (**B**); and *Squalus blainville* (**C**).

**Figure 3 biology-12-01150-f003:**
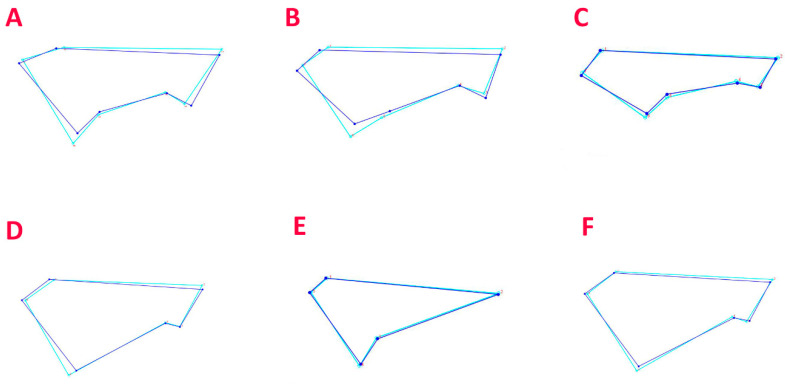
Wire-framed graphs produced by the discriminant function analysis of *C. uyato* (**A**), *D. licha* (**B**), *E. spinax* (**C**), *O. centrina* (**D**), and *S. blainville* (**E**), where caudal fins grouped by size group (blue = small, light blue = large); wire-framed graph of *O. centrina* caudal fins grouped by sex (**F**) (blue = males, light blue = females).

**Figure 4 biology-12-01150-f004:**
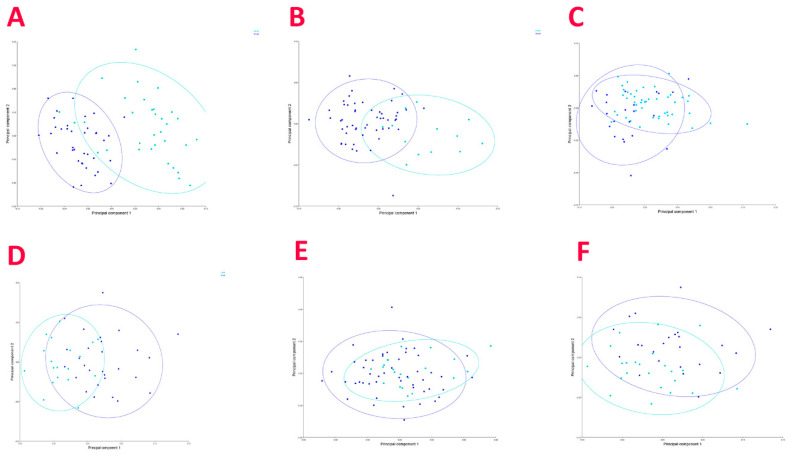
Principal component analysis performed on the caudal fin shape of *C. uyato* (**A**), *D. licha* (**B**), *E. spinax* (**C**), *O. centrina* (**D**), and *S. blainville* (**E**), where caudal fins grouped by size group (blue = small, light blue = large); PCA performed on *O. centrina* caudal fins grouped by sex (**F**) (blue = males, light blue = females).

**Table 1 biology-12-01150-t001:** Acronym and brief description of each body measurement recorded in the investigated species.

Acronym	Description	Acronym	Description
TL	Total length	MOW	Mouth width
FL	Fork length	D1A	First dorsal fin anterior margin
PD1	Pre-first dorsal fin length	D1B	First dorsal fin base length
PD2	Pre-second dorsal fin length	D1B^I^	First dorsal fin spine to inner margin
PG1	Prebranchial length	DIH	First dorsal fin height
IDS	Interdorsal space	D1I	First dorsal fin inner margin
DCS	Dorsal caudal fin space	D2A	Second dorsal fin anterior margin
CDM	Dorsal caudal fin margin	D2B	Second dorsal fin base length
PCA	Pelvic fin caudal fin space	D2B^I^	Second dorsal fin spine to inner margin
CPV	Preventral caudal fin margin	D2H	Second dorsal fin height
HDL	Head length	D2I	Second dorsal fin inner margin
PP2	Prepelvic fin length	P1A	Pectoral fin anterior margin
SVL	Snout-vent length	P1B	Pectoral fin base
POB	Preorbital length	P1H	Pectoral fin height
INO	Interorbital length	P1I	Pectoral fin inner margin
PSP	Prespiracular length	P2A	Pelvic fin anterior margin
PN	Prenostril length	P2B	Pelvic fin base
POR	Preoral length	P2H	Pelvic fin height
INW	Internostril space		

**Table 2 biology-12-01150-t002:** Summary of the sample composition of specimens used for body morphometric analysis, grouped by sex (M—males, F—females) and size group (S—small, L—large).

		Sex		Size Group		Total
Species		M	F	S	L	
*D. licha*	Number	46	57	81	22	103
	Size range	30.64–87.39	30.27–109.44	30.27–58.39	68.8–109.44	30.27–109.44
	Mean TL	45.72	49.45	36.43	87.27	47.29
	Standard deviation	18.77	24.64	4.24	11.48	21.90
*C. uyato*	Number	55	40	48	47	95
	Size range	37.95–89.84	40.42–104.00	37.95–53.97	55.90–104.00	37.95–104
	Mean TL	62.93	65.41	46.45	81.87	63.97
	Standard deviation	17.08	22.76	4.01	10.99	19.60
*S. blainville*	Number	51	29	56	24	80
	Size range	28.7–59.50	22.00–79.20	22.00–39.90	40.50–79.20	22.00–79.20
	Mean TL	39.39	41.69	34.91	52.62	40.22
	Standard deviation	7.88	13.48	3.33	10.23	10.24
*E. spinax*	Number	50	74	60	64	124
	Size range	11.99–31.62	10.61–42.30	10.61–24.83	25.12–42.30	10.61–42.3
	Mean TL	22.39	25.39	17.55	30.40	24.18
	Standard deviation	6.13	8.57	4.02	4.72	7.79
*O. centrina*	Number	22	25	28	19	47
	Size range	22.97–63.15	24.57–74.61	22.97–54.48	55.02–74.61	22.97–74.61
	Mean TL	48.61	57.01	44.82	65.25	53.08
	Standard deviation	11.80	13.92	10.56	6.07	13.51

**Table 3 biology-12-01150-t003:** Summary of the CAP results for sex (above; F = females, M = males) and size (below; S = small, L = large), including correlation and correlation-squared (Corr. Sq.) values, the overall percentage of misclassification, and information on how each a priori classified specimen (Orig. Group) was grouped by the analysis.

Sex as Factor								
					Computed Group			
Species	Correlation	Corr. Sq.	Misclassification Error (%):	Orig. Group	F	M	Total	%Correct
*Centrophorus uyato*	0.5744	0.3299	36.842	F	27	13	40	67.500
				M	22	33	55	60.000
*Dalatias licha*	0.6361	0.4047	39.806	F	35	22	57	61.404
				M	19	27	46	58.696
*Etmopterus spinax*	0.4129	0.1705	33.065	F	51	23	74	68.919
				M	18	32	50	64.000
*Oxynotus centrina*	0.4531	0.2053	34.043	F	18	7	25	72.000
				M	9	13	22	59.091
*Squalus blainville*	0.685	0.4692	25.000	M	40	11	51	78.431
				F	9	20	29	68.966
**Size as Factor**								
					**Computed Group**			
	**Correlation**	**Corr. Sq.**	**Misclassification Error (%):**	**Orig. Group**	**Large**	**Small**	**Total**	**%Correct**
*Centrophorus uyato*	0.8459	0.7155	6.316	Large	42	5	47	89.362
				Small	1	47	48	97.917
*Dalatias licha*	0.894	0.7993	0.971	large	22	0	22	100.000
				small	1	80	81	98.765
*Etmopterus spinax*	0.7817	0.611	16.129	Large	55	9	64	85.938
				Small	11	49	60	81.667
*Oxynotus centrina*	0.8034	0.6454	12.766	Large	16	3	19	84.211
				Small	3	25	28	89.286
*Squalus blainville*	0.756	0.5716	12.500	Small	49	7	56	87.500
				Large	3	21	24	87.500

**Table 4 biology-12-01150-t004:** Summary statistics regarding the sample composition of specimens used for geometric morphometric analysis of the caudal fin, grouped by sex (M—males, F—females) and size group (S—small, L—large).

		Sex		Size Group		Total
Species		M	F	S	L	
*D. licha*	Number	24	42	53	13	66
	Size range	33.90–89.00	31.50–104.30	31.39–58.39	68.80–109.44	31.50–109.30
	Mean TL	42.61	51.46	36.56	88.71	47.51
	Standard deviation	16.55	25.17	4.13	13.97	22.06
*C. uyato*	Number	45	25	37	33	70
	Size range	41.58–89.84	38.70–104.00	38.70–54.30	55.90–104.00	38.70–104.00
	Mean TL	63.96	60.14	46.68	80.44	62.59
	Standard deviation	17.00	21.77	3.80	11.05	18.77
*S. blainville*	Number	45	25	49	21	70
	Size range	28.70–59.50	22.00–79.20	22.00–39.90	40.50–79.20	22.00–79.20
	Mean TL	39.80	41.66	34.92	53.40	40.47
	Standard deviation	8.22	13.99	3.47	10.32	10.59
*E. spinax*	Number	27	43	23	47	70
	Size range	13.20–39.00	11.60–43.5	11.60–24.60	25.5–43.5	11.60–43.50
	Mean TL	24.89	28.85	18.42	31.68	27.32
	Standard deviation	6.52	8.27	3.72	5.15	7.84
*O. centrina*	Number	22	25	28	19	47
	Size range	22.97–63.15	24.57–74.61	22.97–54.48	55.02–74.61	22.97–74.61
	Mean TL	48.61	57.01	44.82	65.25	53.08
	Standard deviation	11.80	13.92	10.56	6.07	13.51

## Data Availability

Raw data belong to the University of Cagliari, and can be requested to the corresponding author with the permission of University of Cagliari.

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
