# Peer review of "Ontogenetic Shifts in Body Morphology of Demersal Sharks’ Species (Order: Squaliformes) Inhabiting the Western-Central Mediterranean Sea, with Implications for Their Bio-Ecological Role"

_biology, 2023, doi:10.3390/biology12081150_

Round 1

Reviewer 1 Report

The manuscript by Bellodi et al. presents interesting new data on the ontogenetic paths of several species of common and not-so-common demersal sharks of the central-western Mediterranean Basin. The results reported therein are noteworthy, as they suggest that while males and females of the analysed species do not significantly differ from each other in terms of (total) body morphology, small (=immature) and large (=late juveniles and adult) individuals do so, with the former being generally more elongated than the latter. Ultimately, these results may prove important for addressing new research on the biology of these deep-water forms, which in turn may prove relevant for their conservation.

I find Bellodi et al.'s manuscript to be acceptable for publication on Biology pending some revisions, in particular:

1) The English text is by and large intelligible, but somewhat sub-standard. Many trivial grammatical errors are present, and sometimes the phrasing appears as somewhat awkward. I have tried and re-write parts of the abstract and Introduction to provide some sort of guidance toward a more fluent English text, but I must admit I am not a native English speaker myself! Anyway, the authors should perform a thorough check of their manuscript before re-submitting it to Biology for further consideration.

2) The authors state that "the most important measurements in distinguishing these two groups [i.e., small and large sharks] were those related to body lengths indicating the large individuals as more elongated". They do also indicate some measurements that may be especially responsible for the observed differences. That said, it would be very important to provide more precise data on how the single measurements contribute to differences - e.g. by means of a devoted table. This would be very important in my opinion, and in fact mandatory in my opinion. For example, how each measurement contribute to the main principal components?

3) Other minor suggestions and possible edits are detailed in the attached PDF file.

Best, 
the reviewer

Please see Point 1 above.

Author Response

REV#1:

The manuscript by Bellodi et al. presents interesting new data on the ontogenetic paths of several species of common and not-so-common demersal sharks of the central-western Mediterranean Basin. The results reported therein are noteworthy, as they suggest that while males and females of the analysed species do not significantly differ from each other in terms of (total) body morphology, small (=immature) and large (=late juveniles and adult) individuals do so, with the former being generally more elongated than the latter. Ultimately, these results may prove important for addressing new research on the biology of these deep-water forms, which in turn may prove relevant for their conservation.

I find Bellodi et al.'s manuscript to be acceptable for publication on Biology pending some revisions, in particular:

1) The English text is by and large intelligible, but somewhat sub-standard. Many trivial grammatical errors are present, and sometimes the phrasing appears as somewhat awkward. I have tried and re-write parts of the abstract and Introduction to provide some sort of guidance toward a more fluent English text, but I must admit I am not a native English speaker myself! Anyway, the authors should perform a thorough check of their manuscript before re-submitting it to Biology for further consideration.

Answer: We are grateful to the reviewer for the work done on our manuscript. We truly believe that her/his efforts have greatly contributed to increase the overall MS quality. We really appreciated the suggestion regarding the language  and the text structure. We are pleased to inform the reviewer that we carefully followed  all the suggestion she/he provided in the PDF file.

2) The authors state that "the most important measurements in distinguishing these two groups [i.e., small and large sharks] were those related to body lengths indicating the large individuals as more elongated". They do also indicate some measurements that may be especially responsible for the observed differences. That said, it would be very important to provide more precise data on how the single measurements contribute to differences - e.g. by means of a devoted table. This would be very important in my opinion, and in fact mandatory in my opinion. For example, how each measurement contribute to the main principal components?

Answer: We plainly agree. The SIMPER analysis was indeed carried out in order to unravel the contribution of the single measurements to the overall differences between groups. All the simper values are reported in the Supplementary table 1. But we noticed that we failed to place the appropriate reference in the text to such table. Added.

3) Other minor suggestions and possible edits are detailed in the attached PDF file.

Answer: We appreciated the efforts done in helping us to improve the overall quality of our MS. We are pleased to inform the reviewer that we dealt with every comment and suggestion she/he indicated in the pdf. Specific answers to each comment are reported below  

Line 41: ?? unclear

Answer: ok, rephrased (L.47)

Line 46: Does 'their' refers to the sharks? If so, please rephrase - otherwise, 'their' would refer to stakeholders and programs...

Answer: Yes we meant “sharks”, but it was unclear. Thank for noticing it. Rephrased (L.51)

Line 66: I don't think that all that relates to sexual dimorphism could be related to the mating behaviour.

Answer: Agreed, we don’t think it either, but in this case we were simply reporting information from existing literature. Anyway, the message was unclear so the sentence has been modified (L.71-73)

Line 81: ?? Sea of Sardinia?

Answer: actually accepted name is Sardinian Seas, which is the name reported for GSA11 by FAO and GFCM reports

Line 93: Here I stop the word-by-word revision of your manuscript, and turn to leave more general comments.

Answer: we are grateful for the efforts produced in helping us to improve our MS quality

Line 102: ??

Answer: Modified (L107).

Line 103: It would be crucial to know whether similar objectives were used.

Answer: true, information added (L109).

Line 118: Please explain acronyms at first usage

Answer: true, information added (L.112-113)

Line 250: Please provide some percentages, as you did for the whole body analyses.

Answer: Ok, information added (L. 257).

Line 264: This statement seem to contradict plainly the previous sentence.

Answer: In some way it was meant to. The previous sentence was related to other studies conducted on other species. Anyway, the sentence was unclear, it was rephrased (L.265-268)

Line 273: It would be very important to provide more precise data on how the single measurements contribute to differences - e.g. by means of a devoted table. This would be very important in my opinion, and in fact mandatory in my opinion. For example, how each measurement contribute to the main principal components?

Answer: We agree. The SIMPER analysis was indeed carried out in order to unravel the contribution of the single measurements to the overall differences between groups. All the simper values are reported in the Supplementary table 1. But we noticed that we failed to place here in the text the reference to such table. Added (277-278).

Reviewer 2 Report

In the overall, the paper is well structured although not that original or disruptive and the results are quite the expected, at least for the sizes. Regarding the general absence of differences between sexes, the authors fail to explain what they even state in the manuscript. In fact, considering the known mating behaviour, there should be differences between sexes as already observed for other species.

Nevertheless, the paper should be improved with a review of the language and structure; there are phrases that are just to long (e.g., in discussion, lines 270-275, there is only one phrase), and the issue focused or its complexity does not justify it.

In the overall, the paper is well structured although not that original or disruptive and the results are quite the expected, at least for the sizes. Regarding the general absence of differences between sexes, the authors fail to explain what they even state in the manuscript. In fact, considering the known mating behaviour, there should be differences between sexes as already observed for other species.

Nevertheless, the paper should be improved with a review of the language and structure; there are phrases that are just to long (e.g., in discussion, lines 270-275, there is only one phrase), and the issue focused or its complexity does not justify it.

Author Response

In the overall, the paper is well structured although not that original or disruptive and the results are quite the expected, at least for the sizes. Regarding the general absence of differences between sexes, the authors fail to explain what they even state in the manuscript. In fact, considering the known mating behaviour, there should be differences between sexes as already observed for other species.

Nevertheless, the paper should be improved with a review of the language and structure; there are phrases that are just to long (e.g., in discussion, lines 270-275, there is only one phrase), and the issue focused or its complexity does not justify it.

Answer: We are grateful to the reviewer for her/his comment. The MS, especially in the language and structure, has been modified in accordance also with the other reviewers’ suggestions.

Reviewer 3 Report

General evaluation

the manuscript analyzes the ntogenetic shifts in body morphology of 5 demersal sharks' in the Western-Central Mediterranean Sea. The topic is very important and the study is approached with suitable techniques and robust analyses. The results are well reported without speculation. I suggest accepting after a few minor revisions.

Specific comments

Materials and Methods

Line 106-107 : Unclear, please rephrase.

Line 270-274 : Unclear please rephrase.

Line 301near (44) insert Bottaro et al 2023 in press for Daliatis licha.

Jaws from the deep: biological and ecological insights on the kitefin shark Dalatias...

Massimiliano Bottaro*,Mauro Sinopoli*,Iacopo Bertocci,Maria Cristina Follesa, Alessandro Cau,Ivan Consalvo, Faustino Scarcelli,Emilio Sperone,Marino Vacchi,Letizia Marsili,Guia Consales,

Frontiers in Marine Science ; DOI: 10.3389/fmars.2023.1155731

Author Response

REV#3

General evaluation

the manuscript analyzes the ontogenetic shifts in body morphology of 5 demersal sharks' in the Western-Central Mediterranean Sea. The topic is very important and the study is approached with suitable techniques and robust analyses. The results are well reported without speculation. I suggest accepting after a few minor revisions.

Answer: We are grateful to the reviewer for the work done on our manuscript. Webelieve that her/his efforts have greatly contributed to increase the overall MS quality.

Specific comments

Materials and Methods

Line 106-107 : Unclear, please rephrase.

Answer: Ok, rephrased (L 112-113).

Line 270-274 : Unclear please rephrase.

Answer: Ok, the sentence has been modified in accordance with the other reviewers’ suggestions (L 277-278).

Line 301near (44) insert Bottaro et al 2023 in press for Daliatis licha. Jaws from the deep: biological and ecological insights on the kitefin shark Dalatias.Massimiliano Bottaro*,Mauro Sinopoli*,Iacopo Bertocci,Maria Cristina Follesa, Alessandro Cau,Ivan Consalvo, Faustino Scarcelli,Emilio Sperone,Marino Vacchi,Letizia Marsili,Guia Consales,Frontiers in Marine Science ; DOI: 10.3389/fmars.2023.1155731

Answer: thank you for the suggestion, reference added (L 302).